# MODEL SELECTION FOR CROSS-LINGUAL TRANSFER USING A LEARNED SCORING FUNCTION

## ABSTRACT

Transformers that are pre-trained on multilingual text corpora, such as, mBERT and XLM-RoBERTa, have achieved impressive cross-lingual transfer learning results. In the zero-shot cross-lingual transfer setting, only English training data is assumed, and the fine-tuned model is evaluated on another target language. No target-language validation data is assumed in this setting, however substantial variance has been observed in target language performance between different fine-tuning runs. Prior work has relied on English validation/development data to select among models that are fine-tuned with different learning rates, number of steps and other hyperparameters, often resulting in suboptimal choices. In this paper, we show that it is possible to select consistently better models when small amounts of annotated data are available in an auxiliary pivot language. We propose a machine learning approach to model selection that uses the fine-tuned model's own internal representations to predict its cross-lingual capabilities. In extensive experiments we find that our approach consistently selects better models than English validation data across five languages and five well-studied NLP tasks, achieving results that are comparable to small amounts of target language development data.[1]

## 1 INTRODUCTION

Pre-trained Transformers (Vaswani et al., 2017; Devlin et al., 2019) have lead to state-of-the-art results on a wide range of NLP tasks, for example, named entity recognition, relation extraction and question answering, often approaching human inter-rater agreement (Joshi et al., 2020a). These models have also been demonstrated to learn effective cross-lingual representations, even without access to parallel text or bilingual lexicons (Wu & Dredze, 2019; Pires et al., 2019). Multilingual pre-trained Transformers, such as mBERT and XLM-RoBERTa (Conneau et al., 2019), support surprisingly effective zero-shot cross-lingual transfer, where training and development data are only assumed in a high resource source language (e.g. English), and performance is evaluated on another target language.

Because no target language annotations are assumed in this setting, source language data is typically used to select among models that are fine-tuned with different hyperparameters and random seeds. However, recent work has shown that English dev accuracy does not always correlate well with target language performance (Keung et al., 2020). In this paper, we propose an alternative strategy for model selection in a zero-shot setting. Our approach, dubbed Learned Model Selection (LMS), learns a function that scores the compatibility between a fine-tuned multilingual transformer, and a target language. The compatibility score is calculated based on features of the multilingual model's learned representations and the target language. A model's features are based on its own internal representations; this is done by aggregating representations over an unlabeled target language text corpus. These model-specific features capture information about how the cross-lingual representations transfer to the target language after fine-tuning on source language data. In addition to model-specific representations, we also make use of learned language embeddings from the `lang2vec` package (Malaviya et al., 2017),[2] which have been shown to encode typological information, for example, whether a language has prepositions or postpositions. To measure compatibility between

---

[1]We will make our code and data available on publication.
[2]https://github.com/antonisa/lang2vec

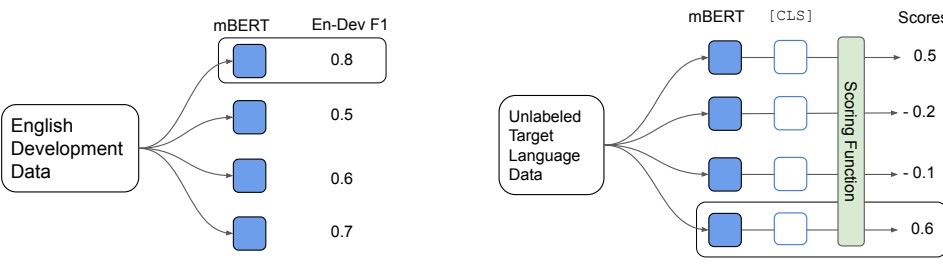

(a) English Development Selection        (b) Learned Model Selection (LMS)

Figure 1: An illustration of our approach to select the best model for zero-shot cross-lingual transfer. (a) Prior works select the best model using source language development data. (b) LMS: A learned function scores fine-tuned models based on their hidden layer representations when encoding unlabeled target language data.

a multilingual model's fine-tuned representations and a target language, the model- and language-specific representations are combined in a bilinear layer. Parameters of the scoring function are optimized to minimize a pairwise ranking loss on a set of held-out models, where the gold ranking is calculated using standard performance metrics, such as accuracy or $F_1$, on a set of pivot languages (not including the target language). Our method assumes training data in English, and small amounts of annotated data in one or more pivot languages (not the target language). This corresponds to a scenario where a new multilingual NLP task needs to be quickly applied to a new language. LMS does not rely on any annotated data in the target language, yet it is effective in learning to predict whether fine-tuned multilingual representations are a good match.

In experiments on five well-studied NLP tasks (part of speech tagging, named entity recognition, question answering, relation extraction and event argument role labeling), we find LMS consistently selects models with better target-language performance than those chosen using English dev data. Appendix A.5 demonstrates that our framework supports multi-task learning, which can be helpful in settings where some target-language annotations are available, but not for the desired task. Finally, we show that LMS generalizes to both mBERT and XLM-RoBERTa in Appendix A.4.

## 2   BACKGROUND: ZERO-SHOT CROSS LINGUAL TRANSFER

The zero-shot setting considered in this paper works as follows. A transformer model is first pre-trained using a standard masked language model objective. The only difference from the monolingual approach to contextual word representations (Peters et al., 2018; Devlin et al., 2019) is the pre-training corpus, which contains text written in multiple languages. For example, mBERT is trained on text written in 104 languages from Wikipedia. After pre-training on a multilingual corpus, the resulting transformer encodes language-independent representations that support surprisingly effective cross-lingual transfer, simply by fine-tuning the pre-trained parameters using English training data. For example, after fine-tuning mBERT using the English portion of the CoNLL Named Entity Recognition dataset, the resulting model can be used to perform inference directly on Spanish text, achieving an $F_1$ score around 75, and outperforming prior work using cross-lingual word embeddings (Xie et al., 2018; Mikolov et al., 2013). A challenge with this approach, however, is that there is a relatively high variance in this performance across training runs. Although the mean $F_1$ score on Spanish is 75, the performance of 60 models fine-tuned with different learning rates and random seeds ranges from around 70 $F_1$ to 78 (Figure 3). In zero-shot learning, no validation/development data is assumed in the target language, motivating the need for a machine learning approach to model selection.

## 3   RANKING MODEL COMPATIBILITY WITH A TARGET LANGUAGE

Given a set of multilingual BERT-based models, $M = m_1, m_2, ..., m_n$ that are fine-tuned on an English training set using different hyperparameters and random seeds, our goal is to select the model that performs the best on a target language, $l_{\text{target}}$. Our approach (LMS) learns to rank a set of models based on two sources of information: (1) the models' own internal representations, and (2) `lang2vec` representations of the target language (Malaviya et al., 2017). By integrating a model's target language representations together with `lang2vec` embeddings, LMS predicts which model will perform best.

We adopt a pairwise approach to learning to rank (Burges et al., 2005; Köppel et al., 2019). The learned ranking is computed using a scoring function, $s(m, l) = f(g_{\text{mBERT}}(m), g_{\text{lang2vec}}(l))$, where $g_{\text{mBERT}}(m)$ is a feature vector for model $m$, which is computed from the model's own hidden state representations, and $g_{\text{lang2vec}}(l)$ is the `lang2vec` representation of language $l$. The model and language features are each passed through a feed-forward neural network and then combined using a bilinear layer to calculate a final score as follows:

$$
\begin{aligned}
s(m, l) &= f(g_{\text{mBERT}}(m), g_{\text{lang2vec}}(l)) \\
&= \text{FFNN}(g_{\text{mBERT}}(m))^T W_{\text{bi}} \text{FFNN}(g_{\text{lang2vec}}(l))
\end{aligned}
$$

Using the above score, we can represent the probability that model $m_i$ performs better than $m_j$ on language $l$:

$$P(m_i \triangleright_l m_j) = \sigma(s(m_i, l) - s(m_j, l))$$

where $\sigma(\cdot)$ is the sigmoid function. To tune the parameters of the scoring function, which include the feed-forward and bilinear layers, we minimize cross-entropy loss:

$$C = \sum_{l \in L \setminus \{l_{\text{target}}\}} \sum_{m_i \in M} \sum_{m_j \in M} -C_{m_i, m_j, l} \tag{1}$$

where

$$C_{m_i, m_j, l} = \mathbb{1}[m_i \triangleright_l m_j] \log P(m_i \triangleright_l m_j) + \mathbb{1}[m_j \triangleright_l m_i] \log P(m_j \triangleright_l m_i)$$

Here $\mathbb{1}[m_j \triangleright_l m_i]$ is an indicator function that has the value 1 if $m_j$ outperforms $m_i$, as evaluated using labeled development data in language $l$.

The first sum in Equation 1 ranges over all languages where development data is available, excluding the target language. After tuning parameters to minimize cross-entropy loss on these languages, the models are ranked based on their scores for the target language, and the highest scoring model, $\hat{m} = \arg\max_m s(m, l_{\text{target}})$, is selected.

## 4   TASKS AND DATASETS

Section 3 presented an approach to selecting a model for zero-shot cross-lingual transfer. To evaluate its effectiveness, a set of fine-tuned models is split into meta-train/dev/test sets. We then use models in the meta-train set to tune parameters of the scoring function, $s(\cdot)$, develop input features with models in the meta-dev set (see §7 for details), and report our main results using models in the meta-test set in Table 2.

We experiment with five well-studied NLP tasks in the zero-shot transfer setting: named entity recognition (NER), part-of-speech (POS) tagging, question answering (QA), relation extraction (RE), and event-argument role labeling (ARL). Labeled training data for each task is assumed in English and trained models are evaluated on the target language. The task/language combinations covered by our experiments are summarized in Table 1.

The following subsections describe the tasks and datasets used in our experiments; more details are presented in Appendix A.1.

|     | ar | bg | da | de | es | fa | hi | hu | it | nl | pt | ro | sk | sl | sv | vi | zh |
|-----|----|----|----|----|----|----|----|----|----|----|----|----|----|----|----|----|----|
| NER |    |    |    | ✓  | ✓  |    |    |    |    | ✓  |    |    |    |    |    |    | ✓  |
| POS | ✓  | ✓  | ✓  | ✓  | ✓  | ✓  |    | ✓  | ✓  | ✓  | ✓  | ✓  | ✓  | ✓  | ✓  |    | ✓  |
| QA  | ✓  |    |    | ✓  | ✓  |    | ✓  |    |    |    |    |    |    |    |    | ✓  | ✓  |
| RE  | ✓  |    |    |    |    |    |    |    |    |    |    |    |    |    |    |    | ✓  |
| ARL | ✓  |    |    |    |    |    |    |    |    |    |    |    |    |    |    |    | ✓  |

Table 1: Seventeen target languages and five tasks used in our experiments. English is used as the source language. ar: Arabic, bg: Bulgarian, da: Danish, de: German, es: Spanish, fa: Persian, hi: Hindi, hu: Hungarian, it: Italian, nl: Dutch, pt: Portuguese, ro: Romanian, sk: Slovak, sl: Slovene, sv: Swedish, vi: Vietnamese, zh: Chinese.

### 4.1 NAMED ENTITY RECOGNITION

We use data from the CoNLL 2002 and 2003 (Sang, 2002; Sang & Meulder, 2003) shared tasks (four target languages: en, de, es, and nl), in addition to a Chinese NER dataset (Levow, 2006), following Wu & Dredze (2019). The data is tagged using the BIO tagging scheme with four types of named entities. We formulate this as a tagging task and add a linear classification layer with a softmax function to obtain word-level predictions. Because the labels are at the word level, but mBERT operates on the subword level, representation of the first subword is used to make word-level predictions, following (Devlin et al., 2019). The CoNLL evaluation script is used to compute F1 scores.

### 4.2 PART-OF-SPEECH TAGGING

A subset of Universal Dependency (UD) Treebanks (v1.4) (Nivre et al., 2016) (fifteen target languages: ar, bg, da, de, es, fa, hu, it, nl, pt, ro, sk, sl, sv, zh) is used, following the setup of Kim et al. (2017). There are eighteen POS tags in total across all languages. We formulate this as a tagging task, following the same approach as described for named entity recognition, and report the accuracy of the predicted POS tags (Acc).

### 4.3 QUESTION ANSWERING

The Multilingual Question Answering (MLQA) dataset (Lewis et al., 2020) is used to evaluate model selection for cross-lingual transfer in the context of question answering. We use the MLQA dataset, consisting of six target languages: ar, de, es, hi, vi, zh. Following the setup of Lewis et al. (2020), models are trained using English data from SQuAD v1.1 (Rajpurkar et al., 2016), and the MLQA English development set is used for the En-Dev baseline. We use the standard BERT approach for question answering (Devlin et al., 2019), where a (question, passage) pair is encoded using special tokens as follows: `[BOS]` Question `[SEP]` Passage `[SEP]`. The classification layer predicts both the start and end index of the answer span from the text passage. The formal evaluation script of MLQA is adopted to compute F1 scores.

### 4.4 RELATION EXTRACTION

Given a pair of entity mentions, RE aims to identify a relation between them and classify its type. We use the ACE 2005 corpus (Walker et al., 2006), which contains two target languages, Arabic and Chinese, using the same preprocessed dataset as Subburathinam et al. (2019). ACE contains eighteen relation types and an additional label indicating no relation (19-way classification). We adopt the best performing model from Soares et al. (2019), [ENTITY MARKERS - ENTITY START], which adds special tokens $[\text{E1}_{\text{start}}], [\text{E1}_{\text{end}}]$ and $[\text{E2}_{\text{start}}], [\text{E2}_{\text{end}}]$ surrounding entity mention pairs in a sentence. The modified sentence is then fed into mBERT and representations of the starting markers $[\text{E1}_{\text{start}}], [\text{E2}_{\text{start}}]$ are concatenated as inputs to a linear classification layer. We follow evaluation protocols established in prior work, using gold entities as inputs to control for differences in performance due to named entity recognition errors (Subburathinam et al., 2019). A predicted relation mention is considered correct if its type and the heads of both entities match the gold data.

### 4.4.1 EVENT-ARGUMENT ROLE LABELING

Event-Argument Role Labeling (ARL) aims to identify event triggers and their arguments in texts. For example, in the sentence *Facebook aquired Instagram*, an event trigger is *aquired*, whose arguments are *Facebook* and *Instagram*, with roles BUYER and SELLER. Again we draw on the ACE 2005 corpus (Walker et al., 2006) (two target languages: ar, zh), using the same preprocessed dataset as Subburathinam et al. (2019). There are thirty-five role labels in ACE. Following a similar approach as was described for relation extraction, we add special markers $[T_{start}]$, $[T_{end}]$ and $[E_{start}]$, $[E_{end}]$ around the trigger and candidate entity mentions. The modified sentence is fed into mBERT, and the marker representations are concatenated as input to a classification layer that predicts the entity's role with respect to the trigger word.

## 5 EXPERIMENTAL DESIGN

For a downstream task with $n$ languages $L : \{l_1, ..., l_n\}$, our goal is to select the model that performs best on a target language, $l_{target} \in L$. We assume the available resources are English training and development data, in addition to a small development set in one or more pivot languages (that do not include the target language). First, a set of mBERT models, $M$, are fine-tuned on an English training set using different hyperparameters and random seeds and shuffled into meta-train/dev/test sets. We then evaluate each model, $m_i$, on the pivot languages' dev sets to calculate a gold ranking, $\triangleright_l$, that is used in the cross-entropy loss (Equation 1). Model-specific features are then extracted from the fine-tuned mBERTs, by feeding unlabeled pivot language text (dev set examples) as input.

**Development and Evaluation** mBERT models in the meta-dev set are used to experiment with different model and language features. Target language dev data was used to experiment with a handful of model and language features, as described in §7. Evaluation is performed using the meta-test set. For each target language, we rank models using the learned scoring function, select the highest scoring model, and report results in Table 2.[3]

### 5.1 BASELINES

En-Dev is our main baseline following standard practice for model selection in the zero-shot setting (Wu & Dredze, 2019; Pires et al., 2019). Model selection with the pivot language dev set is included as another baseline, where the pivot language is selected based on the highest similarity with the target language, as measured using cosine similarity between lang2vec embeddings among candidates in $\{ar, de, es, nl, zh\}$. Also, we compare our results with models selected using 100 sentences annotated in the target language to understand how our approach compares to the more costly alternative of annotating a small amount of target language development data. Finally, an All-Target oracle that picks the best model based on target language development data is adopted. These baselines and oracles are summarized below:

- En-Dev (baseline): choose an mBERT with the English dev set.
- Pivot-Dev (baseline): choose an mBERT with the pivot language dev set.
- 100-Target (oracle): choose an mBERT with 100 target language dev set instances.
- All-Target (oracle): choose an mBERT with the full target language dev set.

### 5.2 HYPERPARAMETERS AND OTHER SETTINGS

To train the scoring function, $s(\cdot)$, we use Adam (Kingma & Ba, 2015), and select the batch size among $\{16, 32, 64, 128\}$, learning rate $\lambda$ among $\{1 \times 10^{-4}, 5 \times 10^{-5}, 1 \times 10^{-5}, 5 \times 10^{-6}, 1 \times 10^{-6}\}$, and train for $\{3\}$ epochs. The scoring function, $s(\cdot)$, contains a 2-layer FFNN with 1024 hidden units and ReLU activation (Glorot et al., 2011). The base cased mBERT has 179M parameters and a vocabulary of around 120k wordpieces. Both the pre-trained transformer layers and task-specific layers are fine-tuned using Adam, with $\beta_1 = 0.9$, $\beta_2 = 0.999$, and an L2 weight decay of 0.01.

---

[3]Evaluation was performed using leave-one-language-out cross-validation over $\{ar, de, es, nl, zh\}$. To evaluate performance on target languages in $\{bg, da, fa, hi, hu, it, pt, ro, sk, sl, sv, vi\}$ for POS and QA, a single LMS was trained using pivot languages in $\{ar, de, es, nl, zh\}$.

Model candidates are fine-tuned with varying learning rates and number of epochs with the following settings: learning rate $\in \{3 \times 10^{-5}, 5 \times 10^{-5}, 7 \times 10^{-5}\}$; number of epochs $\in \{3, 4, 5, 6\}$; batch size $\in \{32\}$; random seeds $\in \{0, 1, ..., 239\}$. 240 mBERT models with different random seeds are fine-tuned with 12 different hyperparameter settings (20 random seeds for each set of hyperparameters), and then split into meta-train/dev/test sets (120/60/60). All models are trained on an RTX 2080 Ti.

## 6 EVALUATION

Below we report model selection results on mBERTs in the meta-test set for each of the five tasks.

| Task | Lang | Ref | En-Dev | Pivot-Dev | LMS | 100-Target | All-Target | # All-Target |
|---|---|---|---|---|---|---|---|---|
| NER ($F_1$) | de | 69.56 | 69.89 | **70.72** (nl) | 69.89 | 66.74 | 72.07 | 2867 |
| | es | 74.96 | 74.61 | 73.10 (nl) | **75.74** | 75.74 | 75.73 | 1915 |
| | nl | 77.57 | 78.74 | **79.27** (de) | 78.85 | 78.67 | 80.26 | 2895 |
| | zh | 51.90 | 54.90 | 52.99 (de) | **55.05** | 55.36 | 56.88 | 4499 |
| POS (Acc) | ar | - | 49.66 | 50.28 (de) | **51.61** | 50.56 | 52.70 | 786 |
| | de | 89.8 | 89.27 | 88.75 (nl) | **89.77** | 89.45 | 89.98 | 799 |
| | es | 85.2 | 84.84 | 85.26 (nl) | **85.56** | 84.82 | 85.11 | 1552 |
| | nl | 75.9 | 75.68 | **75.93** (de) | 75.87 | 75.54 | 76.04 | 349 |
| | zh | - | 66.94 | 66.90 (de) | **68.00** | 67.25 | 68.84 | 500 |
| | bg | 87.4 | 87.13 | 87.05 (es) | **87.92** | 87.95 | 87.95 | 1115 |
| | da | 88.3 | 88.64 | 88.87 (nl) | **88.94** | 88.64 | 89.20 | 322 |
| | fa | 72.8 | 71.64 | 71.63 (es) | **73.63** | 73.63 | 73.75 | 599 |
| | hu | 83.2 | 82.55 | 82.05 (de) | **83.26** | 83.26 | 83.11 | 179 |
| | it | 84.7 | 84.47 | 84.89 (es) | **85.23** | 85.37 | 85.84 | 489 |
| | pt | 82.1 | 81.80 | 81.88 (es) | **82.16** | 81.82 | 82.18 | 271 |
| | ro | 84.7 | 83.83 | 84.19 (es) | **84.71** | 84.43 | 85.37 | 1191 |
| | sk | 83.6 | 83.73 | 83.65 (es) | **84.23** | 83.65 | 84.80 | 1060 |
| | sl | 84.2 | 84.49 | 83.48 (es) | **85.16** | 83.82 | 85.53 | 735 |
| | sv | 91.3 | 91.39 | **91.83** (nl) | 91.66 | 91.33 | 91.76 | 504 |
| QA ($F_1$) | ar | 45.7 | 47.72 | **49.36** (de) | 49.26 | 49.36 | 49.36 | 517 |
| | de | 57.9 | 55.27 | 55.83 (ar) | **55.86** | 57.12 | 55.83 | 512 |
| | es | 64.3 | 64.92 | 64.74 (ar) | **64.95** | 64.51 | 65.08 | 500 |
| | zh | 57.5 | 58.03 | 58.09 (de) | **58.11** | 58.14 | 58.38 | 504 |
| | hi | 43.8 | 39.09 | 42.05 (es) | **42.43** | 38.85 | 42.89 | 507 |
| | vi | 57.1 | 57.36 | 56.88 (ar) | **58.21** | 59.12 | 58.12 | 511 |
| RE ($F_1$) | ar | 39.43 | 36.10 | 35.35 (zh) | **39.54** | 34.68 | 41.92 | 4482 |
| | zh | 32.74 | 67.68 | 67.43 (ar) | **70.75** | 68.20 | 69.13 | 7096 |
| ARL ($F_1$) | ar | 16.48 | 44.11 | **48.08** (zh) | 47.08 | 44.11 | 47.15 | 1221 |
| | zh | 23.49 | 60.96 | 61.26 (ar) | **62.05** | 62.52 | 63.81 | 2226 |

Table 2: Model scores selected based on LMS for NER, POS, QA, RE, and ARL. En-Dev / Pivot-Dev / 100-Target / All-Target: model selection based on the highest F1 of English dev set / Pivot language dev set (pivot language in bracket) / 100 target language dev set examples / target language dev set. LMS: model selection based on the highest scores for the target language: $\arg\max_m s(m, l_{\text{target}})$; "# All-Target" is the number of labeled target-language sentences used for model selection in the All-Target oracle.

**NER** As illustrated in Table 2, our method selects models with a higher F1 score than En-Dev except in the case of German (de). Besides, it outperforms model selection using small amounts of target-language annotations (100-Target) on Dutch (nl) and selects a model that performs as well on Spanish (es). On average, LMS achieves 0.86 point increase in F1 score relative to Pivot-Dev. We use (Wu & Dredze, 2019) as references for zero-shot cross-lingual transfer with mBERT.

**POS** Table 2 presents POS accuracies on the test set, using various approaches to model selection for the fifteen target languages. LMS outperforms En-Dev and Pivot-Dev except in the case of Swedish (sv) and Dutch (nl). Interestingly, model selection for Italian with Spanish dev set does not outperform LMS. We use (Wu & Dredze, 2019) as references for zero-shot cross-lingual transfer with mBERT.

**QA** Our method selects a model with higher F1 across all languages compared with En-Dev, although we find that Pivot-Dev performs slightly better on Arabic (ar). We use (Lewis et al., 2020) as references for zero-shot cross-lingual transfer with mBERT.

**ARL and RE** In Table 2, our method selects models with higher F1 scores compared to En-Dev. It also outperforms 100-Target across both languages. We hypothesize this is because 100 target-language examples is not sufficient for effective model selection, as the dataset contains a large proportion of negative examples (no relation). Also, RE and ARL have large label sizes (18 and 35) so a random sample of 100 instances might not cover every label. In contrast, the full dev set contains thousands of examples. We use $GCN_{ReImp}$ as a reference in the imbalanced dataset (see Appendix A.1 for details). Reference models were selected using the English dev set.

**Model Score Distributions** Figure 2 visualizes the En-Dev and LMS results on the test set in the context of the score distributions of the 60 models in the meta-test set, using kernel density estimation. English development data tends to select models that perform only slightly better than average, whereas LMS does significantly better. Similar visualizations for NER, POS, and QA are presented in Appendix A.2. In addition to the model score, we run a statistical analysis of the results for POS and QA in Appendix A.3. LMS is statistically significantly higher than En-Dev while Pivot-Dev fails in three target languages.

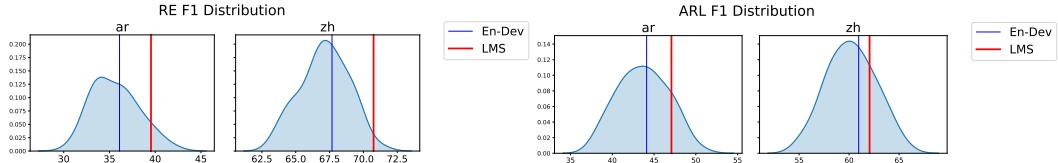

Figure 2: Model F1 score distributions for RE and ARL. Red line: LMS and blue line: En-Dev. X-axis is F1 score. Selecting models with LMS achieve better results compared to En-Dev.

# 7 ANALYSIS

In Section 6, we empirically demonstrated that our learned scoring function, $s(\cdot)$, consistently selects better models than the standard approach (En-Dev), and is comparable to small amounts of labeled target language data. Section 7 presents additional analysis of our approach, exploring the impact of various modeling choices with $\{ar, de, es, nl, zh\}$. In addition, analysis of generalization beyond mBERT, across tasks capability, and size of training are present in Appendices A.4, A.5, and A.6.

## 7.1 MODEL AND LANGUAGE FEATURES

This section explores the impact of different choices for model and language representations for LMS. Four types of model features and two language embeddings are explored. We start by delineating possible choices for representations, then describe the details of our experiments, results, and the final choices used in §6.

Four model-specific features are described below. Note `[CLS]` vectors are extracted from mBERT by feeding unlabeled text as input.

- `[Eng]`: Averaged `[CLS]` vectors computed over an unlabeled English text corpus are used for both training and testing.[4]
- `[Pivot]`: During training, `[CLS]` vectors are averaged over an unlabeled text corpus in the pivot language. At test time, `[CLS]` embeddings are averaged over an unlabeled corpus in the target language. We use the target-language development set (ignoring labels) for this purpose in our experiments.
- `[Target]`: `[CLS]` vectors are averaged over a text corpus in the target language (for both training and testing).

---

[4]In our experiments, sentences in the English dev set are used for this purpose (ignoring the labels).

- Fusion: A linear combination of the above features. Weights on each representation are learned during training.

Two types of language embeddings are examined.

- `lang2vec`: 512-dimensional vectors learned by a neural network trained for typological prediction (Malaviya et al., 2017).
- `syntax`: 103-dimensional binary vectors, which capture syntax features from the URIEL knowledge base (Littell et al., 2017).

First, we determine the choice of model-specific features by averaging performance across both language embeddings. Table 3 reports averaged evaluation metrics for each model-specific representation across all target languages with En-Dev as a baseline.

Averaged evaluation metrics across all target languages for each language embedding are reported in Table 4. In addition to evaluating the effectiveness of each language embedding, we also experimented with a variant of our scoring function that does not include any language embeddings as input. Results are reported on mBERT models in the meta-dev set and the target languages' dev sets for all experiments in this section.

| Task | En-Dev | [Eng] | [Pivot] | [Target] | Fusion |
|------|--------|-------|---------|----------|--------|
| NER | 70.45 | 70.64 | 71.18 | **71.87** | 70.66 |
| POS | 74.69 | **75.58** | 75.54 | 75.48 | 75.04 |
| QA | 56.31 | 56.49 | **56.79** | 56.68 | 56.63 |
| RE | 51.81 | 54.92 | **55.57** | 55.56 | 54.57 |
| ARL | 50.98 | 51.99 | 53.74 | 52.31 | **54.69** |
| Avg | 60.85 | 61.92 | **62.60** | 62.38 | 62.32 |

Table 3: Model-specific feature analysis. We use mBERT models in the meta-dev set for analysis. Each number represents average of scores across all the target languages in a particular task.

| Task | En-Dev | lang2vec | syntax | None |
|------|--------|----------|--------|------|
| NER | 70.45 | **71.37** | 70.98 | 70.08 |
| POS | 74.69 | **75.72** | 75.36 | 75.20 |
| QA | 56.31 | **56.81** | 56.77 | 56.49 |
| RE | 51.81 | **55.92** | 55.22 | 52.53 |
| ARL | 50.98 | 53.60 | **53.88** | 53.14 |
| Avg | 60.85 | **62.68** | 62.44 | 61.49 |

Table 4: Language embedding analysis across `lang2vec`, `syntax`, and no language embedding. We use mBERT models in the meta-dev set for analysis. Each number represents average of scores across all the target languages in a particular task.

In Table 3, `[PIVOT]` features achieve top-2 performance in all five tasks. `[Eng]` and `[Target]` achieve mixed results, and the fusion of three features does not effectively incorporate the advantages of each representation, except in the case of ARL. Table 4 shows that `lang2vec` outperforms `syntax` for all tasks but ARL and also outperforms our approach when language embeddings are not included. Thus, `lang2vec` and `[PIVOT]` are used for all experiments in Section 6.

## 8 RELATED WORK

Recent work has explored hyper-parameter optimization Klein et al. (2019), and model selection for a new task. `task2vec` (Achille et al., 2019) presents a meta-learning approach to selecting a pre-trained feature extractor from a library for a new visual task. More concretely, `task2vec` represents tasks in a vector space and is capable of predicting task similarities and taxonomic relations. It encodes a new task and selects the best feature extractor trained on the most similar task. Unlike `task2vec`, we select a trained model for a specific task, and we represent a trained model with model-specific features on a target language.

MAML (Finn et al., 2017; Rajeswaran et al., 2019) is another approach to meta-learning, pre-training a single model with a meta-loss to initialize a set of parameters that can be quickly fine-tuned for related tasks. Nooralahzadeh et al. (2020) explore the use of MAML in the cross-lingual transfer setting. MAML is designed to support few-shot learning through better initialization of model parameters and does not address the problem of model selection. In contrast, our approach improves model selection in the zero-shot cross-lingual transfer setting.

Most relevant to our work, Xia et al. (2020) use regression methods to predict a model's performance on an NLP task. They formulate this as a regression problem based on features of the task (dataset size, average sentence length, etc.), incorporating a discrete feature to represent the choice of model. In contrast, LMS inspects a model's internal representations, thus it is suitable for predicting which out of a set of fine-tuned models will best transfer to a target language. Also relevant is prior work on learning to select the best language to transfer from Lin et al. (2019).

Finally, we note that there is a need for more NLP research on low-resource languages (Joshi et al., 2020b). Lauscher et al. (2020) present a number of challenges in transferring to languages with few resources using pre-trained transformers. The languages used in our experiments could be considered high-resource, however, our experiments do cover a fairly diverse set of languages, including Arabic and Chinese. We believe that there is still a need for more research on multilingual NLP for high-resource languages as well, as this is not a solved problem. Finally, we note that there are several other prominent benchmarks for evaluating cross-lingual transfer including XTERME (Hu et al., 2020) and XGLUE (Liang et al., 2020), both of which include some datasets used in this work.

## 9    CONCLUSION

In this paper, we presented a machine learning approach to model selection for zero-shot cross-lingual transfer, which is appropriate when small amounts of development data are available in one or more pivot languages, but not in the target language. We showed that our approach improves over the standard practice of model selection using source language development data. Experiments on five well-studied NLP tasks show that by inspecting internal representations, our method consistently selects better models. LMS also achieves comparable results to the more expensive alternative of annotating small amounts of target-language development data.

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

# A APPENDIX

## A.1 IMBALANCED DATA FOR RE AND ARL

In this section, we describe details of the imbalanced dataset for RE and ARL. Table 5 reports the statistics of the dataset and Table 6 summarizes references and baseline results.

The number of non-relation entity pairs or trigger-entity pairs (negative instances) far exceeds positive instances, which negatively affects a model's performance. However, the preprocessed dataset from (Subburathinam et al., 2019) is artificially balanced, which does not reflect a practical setting. Therefore, we create an additional imbalanced dataset using the ACE2005 corpus Walker et al. (2006), which more closely replicates the setting a model will be faced with in a real-world information extraction scenario. First, we shuffle documents into 80%/10%/10% splits for train/dev/test, then extract candidate entity-pairs from each document. For RE, the first approach in (Ye et al., 2019) is adopted to extract negative instances. Negative instances whose entity-type combination has never appeared as a positive example in the training data are filtered out. This results in a positive/negative ratio of 1:8.9 within 80,256 mention pairs for English. We repeat the same process for Chinese/Arabic and end up with a positive/negative ratio of 1:9.4/1:8.9 and 73,082/42,104 instances. For ARL, we create negative instances by pairing each trigger with every entity in a sentence. This leads to a positive/negative ratio of 2.6/2.7/2.9 and 27,823/19,338/14,376 instances in English/Chinese/Arabic. Details on the two datasets are summarized in Table 5.

As a baseline for the imbalanced dataset, we reimplement the Graph Convolutional Network (GCN) model of (Subburathinam et al., 2019) using multilingual embeddings learned by fastText (Bojanowski et al., 2017) on Wikipedia (GCN$_{\text{ReImp}}$). Tables 6 display F1 for zero-shot cross-lingual transfer in both balanced and imbalanced datasets.

| Task | Lang | Train | Dev | Test | Pos/Neg |
|------|------|-------|-----|------|---------|
|      | en   | 63177 | 10218 | 6861 | 1:8.9 |
| RE   | zh   | 57824 | 7096  | 8162 | 1:9.4 |
|      | ar   | 32984 | 4482  | 4638 | 1:8.9 |
|      | en   | 21875 | 3345  | 2603 | 1:2.6 |
| ARL  | zh   | 15095 | 2226  | 2017 | 1:2.7 |
|      | ar   | 11587 | 1221  | 1568 | 1:2.9 |

Table 5: Statistics of the imbalanced dataset. Number of instances and the total positive/negative ratio.

|  | RE ($\mathbf{F_1}$) | | ARL ($\mathbf{F_1}$) | |
|--|------|------|------|------|
|  | ar | zh | ar | zh |
| *Balanced Dataset* | | | | |
| GCN (Subburathinam et al., 2019) | 58.70 | 42.50 | 61.80 | 59.00 |
| GCN$_{\text{ReImp}}$ | 56.10 | 41.70 | 62.08 | 55.10 |
| mBERT | 69.08 | 79.56 | 59.27 | 68.00 |
| *Imbalanced Dataset* | | | | |
| GCN$_{\text{ReImp}}$ | 39.43 | 32.74 | 16.48 | 23.49 |
| *Model Selection (Imbalanced Dataset)* | | | | |
| En-Dev | 36.10 | 67.68 | 44.11 | 60.96 |
| LMS | **39.54** | **70.75** | **47.08** | **62.05** |
| 100-Target | 34.68 | 68.20 | 44.11 | 62.52 |
| All-Target | 41.92 | 69.13 | 47.15 | 63.81 |

Table 6: F1 scores for relation extraction and argument role labeling on the test set. En-Dev/100-Target/All-Target: model selection based on the highest F1 of English dev set/100 target language dev set examples/target language dev set. Ours: model selection based on the highest scores for the target language: $\arg\max_m s(m, l_{\text{target}})$.

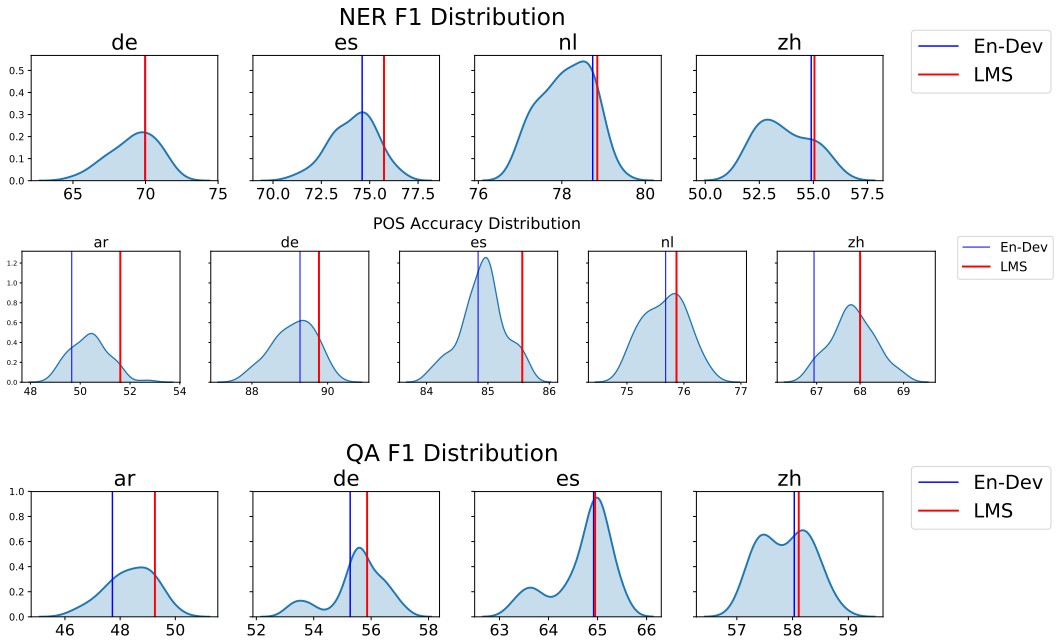

Figure 3: Model score distribution of models in the meta-test set for NER, POS, and QA. LMS (Red) and En-Dev (Blue). X-axis is F1 score for NER and QA and accuracy for POS.

## A.2 MODEL SCORE DISTRIBUTION

Details of model score distribution are explained and analyzed in this section. Distribution figures of NER, POS, and QA are shown in Figure 3.

To better interpret model selection results, we visualize the distribution of model scores in the meta-test set with the kernel density estimate (KDE) plot. KDE represents the data using a continuous probability density curve in one or more dimensions. LMS (Red) and En-Dev (Blue) selection results are marked in the model score distribution. The majority model scores are concentrated at the center of the distribution but the variations are large. For example, Arabic POS accuracy in Figure 3 ranging from 48 to 53, where the majority of models achieve a score around 50. En-Dev fails to select a model from the middle of the distribution, which leads to a score below average models. However, LMS is able to not only select a model better than the En-Dev but also select a model that is better than the majority of candidates. In fact, En-Dev is not a good model selection strategy when it comes to linguistically different languages such as Arabic and Chinese. For Arabic POS, Arabic QA, and Chinese POS, En-Dev selects a model that performs worse than an average model from the candidates (left side of the distribution). This might suggest the model is over-fitting to English data but fails to transfer to a distanced language. Our observations motivate us to develop a better model selection method - LMS.

## A.3 STATISTICAL TEST OF LMS PERFORMANCE

In this section, we present a statistical analysis of model selection results for POS and QA. Table 7 shows LMS on average improve a point of 0.74 relative to and a point of 0.44 relative to Pivot-Dev.

We train a single LMS with pivot languages in $\{ar, de, es, nl, zh\}$ for POS and $\{ar, de, es, zh\}$ for QA, following by testing it on all the target languages. All the results are reported with mean and standard deviation with five runs (different random seeds). A z-test is performed to the differences between LMS/Pivot-Dev and En-Dev.

LMS is statistically significantly ($p$-value $\leq 0.05$) higher than En-Dev baseline across all languages and two tasks while Pivot-Dev fails in three languages. LMS also obtains a lower standard deviation for the model scores except for Swedish (sv) and Vietnamese (vi).

| Methods | POS (Accuracy) | | | | | | | | | | QA (F$_1$) | |
|---|---|---|---|---|---|---|---|---|---|---|---|---|
| | bg | da | fa | hu | it | pt | ro | sk | sl | sv | hi | vi |
| Wu & Dredze (2019) | 87.4 | 88.3 | 72.8 | 83.2 | 84.7 | 82.1 | 84.7 | 83.6 | 84.2 | 91.3 | - | - |
| Lewis et al. (2020) | - | - | - | - | - | - | - | - | - | - | 43.8 | 57.1 |
| *Model Selection* | | | | | | | | | | | | |
| En-Dev | 87.22±0.24 | 88.59±0.17 | 71.58±0.28 | 82.81±0.37 | 84.44±0.45 | 81.87±0.16 | 83.84±0.30 | 83.49±0.42 | 84.26±0.34 | 91.37±0.05 | 39.93±1.45 | 57.18±0.82 |
| Pivot-Dev | 87.35±0.37 (es) | **88.81±0.12** (nl)* | 71.55±1.02 (es) | 82.34±0.27 (de) | 84.68±0.48 (es)* | 82.06±0.23 (es)* | **85.59±0.48** (es)* | **84.09±0.50** (es)* | 84.10±0.57 (es) | **91.63±0.19** (nl)* | 41.40±1.35 (es)* | 57.66±2.03 (ar) |
| LMS | **87.75±0.14*** | 88.74±0.14* | **73.57±0.13*** | **83.28±0.17*** | **85.04±0.19*** | **82.18±0.08*** | 84.74±0.04* | 83.93±0.28* | **84.91±0.24*** | 91.55±0.21* | **42.09±0.91*** | **57.73±1.16*** |
| 100-Target | 87.72±0.50 | 88.74±0.06 | 73.00±0.76 | 83.14±0.17 | 85.14±0.27 | 82.14±0.34 | 84.78±0.47 | 84.07±0.58 | 84.23±0.16 | 91.58±0.23 | 40.04±0.81 | 58.83±0.90 |
| All-Target | 88.09±0.17 | 89.01±0.18 | 73.80±0.31 | 83.18±0.15 | 85.61±0.21 | 82.36±0.17 | 85.39±0.16 | 84.90±0.33 | 85.41±0.61 | 91.73±0.05 | 42.56±0.26 | 59.12±0.92 |
| # of All-Target | 1115 | 322 | 599 | 179 | 489 | 271 | 1191 | 1060 | 735 | 504 | 507 | 511 |

Table 7: Model scores (mean ± sd) selected based on LMS for POS and QA over 5 runs. Bold indicates the best score and underline indicates the second best. ∗ indicates the LMS/Pivot-Dev is statistically significantly ($p$-value ≤ 0.05) higher than En-Dev.

## A.4 Does this Approach Generalize to XLM-RoBERTa?

In Section 6, we showed that our approach consistently selects better fine-tuned models than those chosen using English development data. To test the robustness of our approach with a different multilingual pre-trained transformer, we re-train and evaluate using XLM-RoBERTa-base (Conneau et al., 2019), with the same settings used for mBERT in Section 6 when testing RE and ARL with imbalance dataset.

**RE** In the left section of Table 8, our approach selects a model with a higher F1 score compared to En-Dev in Chinese and on par with En-Dev in Arabic.

**ARL** In the right section of Table 8, our approach selects a model with a higher F1 score compared to En-Dev in Arabic but performs worse on Chinese. Overall, our approach appears to be effective when used with XLM-RoBERTa.

| | RE (F$_1$) | | ARL (F$_1$) | |
|---|---|---|---|---|
| | ar | zh | ar | zh |
| GCN$_{ReImp}$ | 39.43 | 32.74 | 16.48 | 23.49 |
| mBERT | 36.10 | 67.68 | 44.11 | 60.96 |
| *Model Selection* | | | | |
| En-Dev | **40.79** | 64.48 | 50.65 | **62.73** |
| LMS | **40.79** | 65.11 | **52.96** | 61.92 |
| 100-Target | 42.33 | 65.38 | 52.90 | 62.12 |
| All-Target | 44.66 | 65.75 | 53.09 | 62.27 |

Table 8: XLM-RoBERTa experiment: F1 of relation extraction and argument role labeling on the imbalanced dataset. Model selection results are based on XLM-RoBERTa-base models in the meta-test set.

## A.5 Can Multi-task Learning Help?

Our setting does not assume access to the labeled data in the target language for a particular task. However, labeled data in the target language may be available for a relevant auxiliary task, which could help the scoring function learn to better estimate whether a model is a good match for the target language.

To test whether an auxiliary task in the target language might help to select a better model for the target task, we fine-tune sets of mBERT models for ARL and RE. Gold rankings on the models are then computed for each language using the pivot languages' dev sets. Also, another "silver" ranking is computed for each language using the auxiliary task. The scoring function is then trained to rank mBERT models for both tasks. To differentiate the two tasks, a variant of the scoring function, $s(m, l, t)$, which concatenates a randomly initialized task embedding with the language embedding is adopted. Following Section 7, we use model selection results from mBERT models in the meta-dev set and report average target language dev-set results across Chinese and Arabic.

In Table 9, our approach can select a model with a higher F1 score for RE. However, multi-task does not benefit ARL but still outperforms En-Dev. As for future direction, we believe an LMS that is

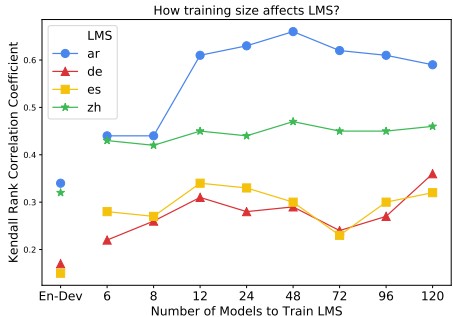

Figure 4: The similarity between En-Dev/LMS ranking and the golden ranking for models in the QA meta-test set.

trained on an auxiliary dataset can be transferred to the target dataset, hence release the requirement of a small amount of pivot language development data in the target dataset.

| Task | En-Dev | ([Pivot], lang2vec) | + Multi-task |
|------|--------|---------------------|--------------|
| RE | 51.81 | 55.92 | **57.31** |
| ARL | 50.98 | **53.60** | 51.99 |

Table 9: Multi-task analysis using additional training data in the target language from another task. We use mBERT models in the meta-dev set for analysis. Model selection is based on the highest scores for the target language and target task: $\arg\max_m s(m, l_{\text{target}}, t_{\text{target}})$

A.6    HOW MANY MODELS DO WE NEED TO TRAIN LMS?

We analyze how practical LMS is in this section. In particular, we aim to explore the question of can the LMS be trained without large amount models. LMS can be trained with only six models.

LMS outperforms the En-Dev in three out of four target languages by training on only six models (Table 10). Varying the number of models in the meta-training set from 6 to 120 for QA, we see LMS performs consistently across a different number of training sizes. It outperforms the En-Dev when trained with 72 models.

To better understand how LMS performs compared to En-Dev, we use Kendall Rank Correlation Coefficient (KRCC) (Kendal, 1938) to measure the similarity of the orderings. KRCC (-1 to 1) between two model rankings will be high when having a similar rank. In Figure 4, KRCC is calculated between En-Dev/LMS and golden model rankings for 60 models in the QA meta-test set. LMS outperforms En-Dev with only six models and the KRCC improves as more models are used to train LMS. It is interesting to see LMS has the highest KRCC in Arabic. We hypothesis this is because of the high variation of F1 score in Arabic - the standard deviation of the Arabic F1 score is 3.3 as opposed to 2.4 and 1.3 in German and Spanish.

| | ar | de | es | zh |
|--|------|------|------|------|
| En-Dev | 47.72 | 55.27 | 64.92 | 58.03 |
| *# of models to learn* LMS | | | | |
| 6 | 48.95 | 55.21 | 65.28 | 58.27 |
| 8 | 48.95 | 55.21 | 65.28 | 58.27 |
| 72 | 48.95 | 56.03 | 64.95 | 58.27 |
| 120 | 49.26 | 55.86 | 64.95 | 58.11 |

Table 10: Model selection results for QA (F1). A varying number of models to train LMS from 6 to 120.

