# OpenReview forum: "Model Selection for Cross-Lingual Transfer using a Learned Scoring Function"
_ICLR.cc/2021/Conference — Reject_

### Official Review · AnonReviewer3 · 2020-10-28
**Interesting Idea but Concern on Experiment**

**Rating:** 7
**Confidence:** 4

**Review:**

## Summary
Research Problem: Zero-shot cross-lingual transfer, with model selection using source language dev set, has high variance with different hyperparameters or even different random seeds.

This paper proposes learning a model selection function using pivot languages dev set. The model selection function is a scoring function, taking the contextual representation and external language embeddings as input. The scoring function is trained with a set of fine-tuned models and its performance ranking in pivot languages dev set. This paper conducts experiments on 5 NLP tasks and cross-lingual transfer from English to 5 target languages, and shows outperforming model selection using the English dev set. It also presents ablation study on the feature of the scoring function and the number of models for training the scoring function.

## Pros
1. This paper presents an interesting approach to learn model selection using model representation and external language embeddings.
2. In the conducted experiments, the proposed method is shown to outperform selecting models using the English dev set.
3. This paper also conducts an ablation study on modeling decisions of the scoring function.

## Cons
1. The modeling assumption seems relatively limited. It assumes only one target language at a time per scoring function (“our goal is to select the model that performs the best on a target language, $l_\text{target}$”). In this experiment where there are multiple target languages of interest, I am assuming it adapts a leave-one-out approach. However, it is not clearly stated. If my understanding is correct, the number of scoring functions increase linearly with the number of target languages. While training the scoring function might be fast (not discussed in the paper), this paper should discuss this assumption in more detail.
2. The experiment's setup is not clearly presented. Aside from the previous example, the experiment's setup is sometimes hard to follow, which makes it relatively hard to assess the claim in the experiment. For example,
- This paper states 240 mBERT models are fine-tuned. But it’s unclear how exactly this count is computed. mBERTs are fine-tuned with 3(learning rate)x4(epochs)x1(batch size)=12 hyperparameter combinations. While it mentioned different random seeds are considered, it is unclear how many seeds are used.
- It is unclear what is the source of unlabeled text.
- It is unclear how many checkpoints each set of hyperparameters produce.
- Most importantly, the meta-test set should come from a set of models with at least **different random seeds**, it’s unclear whether the current split can test the generalization ability of the scoring function.
3. Get back to the original research problem, recall that cross-lingual transfer has high variance. From this paper's main result, it is unclear whether the proposed method has lower variance compared to model selection with source language dev set, as only single performance is reported.

## Questions during rebuttal period
1. In appendix A.3., is a new scoring function trained for XLM-R or is it the same scoring function of mBERT? If the latter is true, I would not refer to it as “generalization”, and the results are quite interesting. If the former is true, more results on other tasks are beneficial as the results so far on XLM-R are mixed.

## Reasons for score
Overall, I am leaning toward rejecting. While I find the idea of learning model selection quite interesting, the experiments presented in this current version cast doubt on the empirical result of this paper. Hopefully the authors can clarify and address my concern in the rebuttal period.

## Minor comment
1. Referring languages other than target language as pivot language seems misleading, as this paper does not assume any similarity between target language and “pivot language”.
2. Referring 100-Target or Full-Target as oracle seems misleading, as the proposed method sometimes even outperforms the “oracle”. The oracle should be the best possible performance in the meta-test set.
3. In table 1, es should have checked instead of nl.

## After rebuttal
Thank you for answering my questions! It address my concern and I revise my recommendation.

- Please clarify in the next version how the checkpoint is selected (e.g. which epoch is selected?) given a product of seed and hyperparameter.

---

> ### Author Response · Authors · 2020-11-23
> **Reply to Reviewer 3**
>
> Thank you for the review and your suggestions. We hope our updated version could address your concerns about the experimental setup.
>
> 1. Yes, this is correct - we used leave-one-language-out cross-validation for ar, de, es, nl, zh.  We have made this clearer in the paper, and also added experiments training the scoring function on the above five languages and evaluating on bg, da, fa, hi, hu, it, pt, ro, sk, sl, sv, vi for UD POS and QA.
>
>
> 2. Experimental details (updated in the new version):
>   - We use 240 random seeds, so each checkpoint has a different random seed.  Each of the 12 hyperparameter settings is evaluated with 20 different random seeds.  We have made this point clearer in the text.
>
>   - The target-language unlabeled data used in our experiments is taken from the development set (ignoring labels).  We have made this clearer in the experiment section of the paper.
>
>
> 3. Variance of the results:
> In the initial draft, we found LMS consistently outperforms En-Dev across five tasks and five diverse languages. We believe this is sufficient to support our claim, but the review does raise a good suggestion to report the variance of the different approaches. In the updated version, we add means and standard deviations for experiments on the 10 new languages for POS and 2 languages for QA (see Appendix 3). The standard deviation of LMS is generally smaller than En-Dev.
>
>
> 4. XLM-R
> Yes, a separate LMS is trained for XLM-R; we have clarified this in the paper.
>
> Finally, we appreciate the reviewer pointing out the typo in Table 1 and we have corrected it.

---

### Official Review · AnonReviewer2 · 2020-10-28
**Relevant contribution, thorough empirical exposition, terribly limited language breadth**

**Rating:** 7
**Confidence:** 3

**Review:**

Summary:
- The most common approach to model selection in zero-shot learning is to use English development data.
- Using English as selection proxy is biased and likely yields suboptimal models.
- The paper proposes a meta-learning based approach to select models by using its own internal representations to predict cross-lingual performance.
- Learned Model Selection (LMS) learns a scoring function that scores how compatible a fine-tuned multilingual transformer is to the target language.
- The target languages are represented by using lang2vec as a reference point.

Strengths:
- Very simple and intuitive model.
- Strong results in comparison to the English dev data baseline.
- Thorough discussion and overall enjoyable writeup.

Weaknesses:
- A disappointing lack of cross-lingual breadth: only six languages in the experiment.
- Especially displeasing that the Universal Dependencies POS test includes only 5 rather resource-rich languages.
- While CoNLL NER data is well-established, there is data for 250+ languages by Pan et al. for NER as well.

Overall:
- I am strongly in favor of the hypothesis and model that the paper proposes and thus I vote accept.
- Would have voted strong accept if not for the underwhelming subset of languages that feature in the experiment.
- I kindly suggest that the final-version revision includes more languages and typological analysis in the experiments.

Questions:
1. How biased is the usage of English development data with respect to language (dis)similarity to target language in these tasks?
2. What would happen if we replaced lang2vec by another representation, e.g., from WALS features or subsets of those?

---

> ### Author Response · Authors · 2020-11-23
> **Reply to Reviewer 2**
>
> Thank you for the helpful suggestions!
>
> In the initial draft of the paper, our experiments showed that LMS outperforms the En-Dev baseline on five tasks and five diverse languages.  We felt this provided sufficient evidence to show the advantage of our approach, but, the review does raise a good point that it is not difficult to extend our experiments to additional languages which are available in the UD POS dataset, and also the question answering dataset.  We have updated the paper to include POS experiments for 10 additional UD languages and question answering for 2 additional languages (Hindi and Vietnamese).
>
> Regarding the choice of the dataset for named entity recognition, we chose CoNLL because it contains high-quality gold standard annotations.  The dataset from Pan et. al. 2017, while interesting due to its linguistic diversity, consists of automatically derived “silver-standard” annotations, so there is a possibility that comparing systems’ output against these noisy labels could introduce biases into the evaluation.  But we agree this could be an interesting avenue for future work.

---

### Official Review · AnonReviewer4 · 2020-10-28
**MODEL SELECTION FOR CROSS-LINGUAL TRANSFER USING A LEARNED SCORING FUNCTION**

**Rating:** 7
**Confidence:** 4

**Review:**

The paper presents an instance of a meta-learning approach, dedicated to performing model selection in a zero-shot setting. This Learned Model Selection (LMS) approach consists of a learned function scoring the compatibility between a fine-tuned multilingual transformer and a target language, without access to any data in target language (hence zero-shot).

The empirical results show that the LMS technique is effective in predicting when a multilingual model’s representations are a good match for the target language, as opposed to a baseline in which the model selection is done using the performance on En dev data. Moreover, it performs similarly with the performance of having access to some small amount of target data (the 100-Target experimental condition), and not very far from the full target data condition (the Full-Target experimental condition).

Interestingly, the authors show that the 512-dimensional vectors learned by a neural network trained for typological prediction (lang2vec) are the most effective features to be used in their approach. This is important because it validates that the degree to which this meta-learning approach works depends on the family of languages for which the task is attempted (the fit between the training-data language family and the target language family).

The paper is well-written, with the experimental section well executed.

Suggestions:
Some of the results are a bit counterintuitive, and I would like to understand a bit better if there are some explanations or intuitions on why that would be; for instance, the LMS approach has better performance on some tasks (RE, ARL) than the 100-Target approach; given that access to the target language should provide inherently some advantages, what is the explanation for that? Is that the 100-Target baseline somehow misuses this advantage?

Somewhat related to the above: I could not see how much data the Full-Target condition is using; seeing that the Full-Target actually performs clearly better on these (RE, ARL) than both LMS and 100-Target, it would be illuminating to understand the threshold N (in terms of data amounts) at which LMS and N-Target would be on-par, especially if that illustrates a correlation between the closeness in the language family for target  vs the value of N.

---

> ### Author Response · Authors · 2020-11-23
> **Reply to Reviewer 4**
>
> Thank you for your review and in particular the suggestion for adding the size of All-Target.
>
> We have updated the paper to include the number of target dev examples as "# All-Target" in Table 2. The statistics of the dev set provide an explanation of the differences between 100-target and all-target and we appreciate your suggestions.
>
> Regarding the question of why LMS performs better than 100-target in some instances (e.g., RE and ARL), we suspect 100 examples might not be sufficient to evaluate the model since the dataset contains a large proportion of negative examples (e.g., no relation). In addition, RE and ARL as a sentence-level classification task have a large label size (18 and 35) so a random 100 examples might not cover every case. In contrast, the full dev set contains thousands of examples.

---

### Official Review · AnonReviewer1 · 2020-10-29
**An interesting problem, a good motivation for the work, but needs more work...**

**Rating:** 6
**Confidence:** 4

**Review:**

=== Update after the revision and the author response ===

I would like to thank the authors for the additional work and effort invested into improving the paper presentation. This has made me increase my score; I still have some doubts regarding the experimental setup (using target dev sets and taking this information for granted), but maybe a high-level question I posed in my review really does go beyond this work. Given that the main premise is 'quick adaptation' to new (and unseen) languages, the inclusion of at least one-two truly low-resource languages would have still been nice to link the motivation with the experimental setup.

===

This paper tackles one interesting problem pertaining to zero-shot cross-lingual transfer, which was observed in previous work. Namely, doing model selection based on English (as the typical source language) dev data often displays suboptimal transfer performance in a wide range of tasks and for a wide range of target languages. The authors set out to tackle this problem by proposing a new approach based on the pairwise learning-to-rank (LTR) framework which combines the multilingual pretrained model's internal representations with some typological knowledge coming from available language vectors (lang2vec vectors are used). The results across five tasks show that doing better model selection (effectively discarding model selection on English dev set) suggest that the LTR-based method does yield improvements in transfer performance. Overall, I see this work as a potentially nice small contribution to the growing area of cross-lingual transfer learning while isolating and motivating the concrete problem in a solid way, but I see a number of ways on how to improve the paper and make this a stronger submission, and I cannot accept it in its current format.

1. While disguising the method as a less resource demanding than the standard approach based on EN dev set, the method is in fact much more resource-demanding. It assumes existence of development data in a sufficient number of target languages (excluding test languages, ofc) which are used to run the LTR training over a number of pretrained mBERT models. Is this a reasonable assumption? How does the amount of languages for which we have development data (and its size) affect the LTR method?

2. Following my previous comment, one pretty obvious baseline/method is missing from the comparisons, if we assume existence of dev data in many languages. For instance, instead of running LTR, why not using development data of a language which is quite similar to the test language, and optimise performance on that 'neighbouring/pivot' language? For instance, if we want to optimise our model to transfer well for Italian NER, and we assume that we possess NER dev data for Spanish, why don't we simply do model selection relying on Spanish NER data (or using Spanish + Catalan or Spanish + Catalan + French NER data if we have dev data for all these languages)? Given that the gains with the LTR approach are still quite small compared to selection based on EN dev, I wonder how this baseline approach would work.

3. Unfortunately, all the experiments are run only on languages which could be considered high-resource languages (e.g., see the work of Joshi et al., ACL 2020 or Lauscher et al.. EMNLP 2020). However, this work in particular should pay particular attention to truly low-resource scenarios, and not limit the evaluations only to a small number of high-resource languages. Low-resource languages are exactly the ones that benefit most from (zero-shot or few-shot) transfer, and the ones where a more careful model selection should really make the difference. However, the paper does not offer any experiments in those setups. Perhaps running some additional experiments with evaluation data assembled in the recent XTREME and XGLUE benchmarks could be insightful?

4. Model-specific feature representation - given that there are so many different ways (e.g., mean/max pooling, using [CLS], using only first-occurring during subwords, layerwise averaging) to extract model-specific feature representations, the authors should provide more justification on choosing these particular feature representations? Have they explored a wider set of options here?

5. Another question relates to a higher-level relationship between model selection versus in-task fine-tuning - related to my comments 1) and 2) above, if we assume existence of some pivot language dev data, the question is whether it is better to use the data for model selection or simply do task-specific fine-tuning on the dev data from related languages to increase the model capability to deal with the final test language? Do the authors have any insights here?

6. I find the use of term 'meta-learning' a bit misleading in this context, as it does not fully relate to the standard research domain of 'meta-learning', and it might even mislead the reader. The work is not strictly 'meta-learning'.


Minor:
- Is there a way to automatically define the number of models to train LMS and optimise their differences so that we do not use redundant models?
- Given that there are multiple possible approaches to choosing a suitable model for transfer, can the authors provide additional discussion and justification for their chosen approach? The problem is indeed most intuitively tackled as a learning-to-rank problem, but there are also many different flavours of LTR. Why this particular implementation?
- I would suggest to cite this highly relevant paper, which actually resembles the main ideas from this paper quite a bit, but instead of choosing between many models from the same language, it chooses models from different languages (1 model in each language = many models, but 1 per language) to optimise for transfer performance: https://arxiv.org/pdf/1905.12688.pdf

---

> ### Author Response · Authors · 2020-11-23
> **Reply to Reviewer 1**
>
> Thank you for the suggestions to improve our paper!  Responses to each point are listed below.
>
> 1. As correctly noted in the review, our method assumes training data in English, and small amounts of annotated development data in one or more pivot languages (not in the target language).  This corresponds to a scenario where a multilingual NLP system needs to be quickly adapted to a new language.  We added a few sentences to the abstract and introduction of the paper to more clearly state assumptions about available resources.
>
>
> 2. Pivot language baseline: Good idea - we have added this to Table 2 (Pivot-Dev).
>
>
> 3. There is a need for more NLP research on low-resource languages, but we would argue that there is also still a need for work on multilingual NLP for high-resource languages, as this is not a solved problem.  The choice of languages used in our work is largely driven by the availability of high-quality multilingual NLP datasets to facilitate evaluation.  We would like to point out that our experiments do include a fairly diverse set of languages, including Chinese and Arabic - we have also added experiments on Hindi and Vietnamese in the revision.  We agree that experimenting with our method on more low-resource languages is an interesting direction for future work.
>
>
> 4. We did include experiments on different choices of feature representations in section 7.1.  These include different choices of unlabeled corpora for computing model representations and we also compared the use of lang2vec and syntax features from the URIEL knowledge base.  We only tried [CLS] features, as these are commonly used for many classification tasks using BERT, but we agree that the other methods of pooling suggested could be interesting to try in future work.
>
>
> 5. The review raises a good question about what is the most effective way to make use of a small amount of pivot language development data (further fine-tuning? or model selection?)  One could consider an approach where the model is trained on both English and a small amount of pivot language dev data, but this doesn’t solve the problem of model selection which is the main focus of our work.
>
>
> 6. We believe that our approach could be considered an instance of meta-learning, but the review raises a good point that the use of this term could be confusing, as we are not contributing a general method for meta-learning that applies to many problems / domains (as is the case for a lot of recent work related to MAML, etc.)  Instead, our contribution is a better way to do model selection for cross-lingual transfer learning in a situation where small amounts of development data are available in a few pivot languages.  We have updated the text of the paper to avoid this potential confusion.
>
> Finally, thank you for the helpful references, which we have used to improve the related work section.

---

### Author Response · Authors · 2020-11-23
**General reply to reviewers**

We would like to thank all reviewers for their thoughtful comments. We are happy to see that reviewers had positive things to say about the paper: **"I am strongly in favor of the hypothesis and model that the paper proposes and thus I vote accept." (R2)**, the paper is **"isolating and motivating the concrete problem in a solid way" (R1)**, and **“The paper is well-written, with the experimental section well executed. "(R4)**

We are also grateful to the reviewers for several constructive suggestions. Inspired by their feedback, additional experiments have been added in the latest revision.  For example, per R1's and R2's suggestions, we add experiments on 10 additional languages for UD POS and 2 languages (Vietnamese and Hindi) for QA, finding that our proposed method outperforms the English dev baseline across these additional languages. We also conducted experiments comparing the variance of the results across model selection methods as suggested by R3, further validating the usefulness of our proposed approach.  Finally, we have added an additional baseline, Pivot-Dev, per R1’s suggestion.  We updated the draft to add these experiments and incorporate other improvements suggested by the reviewers.

Below, we respond with more detail to each review in corresponding threads.

---

### Decision · Program_Chairs · 2021-01-07
**Final Decision**

**Decision:**

Reject

**Comment:**

While the paper has merits, the experiments are lacking in important respects: I agree with Reviewer 1 that it is a serious problem that the approach is not evaluated on truly low-resource languages - since a significant pivot-to-target language bias is to be expected (as also suggested by Reviewer 2). I also agree with the sentiment that the work is not properly baselined, without considering alternative ways of using the pivot language development data. I also agree with Reviewer 3 that the 1:1 assumption is limiting, given that multi-source transfer has been de facto standard since 2011 (see, e.g., work by McDonald, Søgaard, Cohen, etc.). I’m also a little worried about using dev data for unlabelled data, since this is data from the exact same sample as the test data. In practice, dev data will be biased, and artificially removing this bias will lead to overly optimistic results.

---

> ### Author Response · Authors · 2021-03-26
> **Summary**
>
> The final scores for the paper are 6,7,7,7 which is in the top 7.5% of papers in the conference in terms of review scores. However, the AC decided to unilaterally reject the paper for reasons that we respectfully disagree with. The paper does not claim that our methods can address low-resource languages.